# Resistance of Primary Photosynthesis to Photoinhibition in Antarctic Lichen *Xanthoria elegans*: Photoprotective Mechanisms Activated during a Short Period of High Light Stress

**DOI:** 10.3390/plants12122259

**Published:** 2023-06-09

**Authors:** Miloš Barták, Josef Hájek, Mehmet Gökhan Halıcı, Michaela Bednaříková, Angelica Casanova-Katny, Peter Váczi, Anton Puhovkin, Kumud Bandhu Mishra, Davide Giordano

**Affiliations:** 1Laboratory of Photosynthetic Processes, Department of Experimental Biology, Faculty of Science, Masaryk University, Kamenice 5, 625 00 Brno, Czech Republic; jhajek@sci.muni.cz (J.H.); bednarikova.mm@gmail.com (M.B.); vaczi@sci.muni.cz (P.V.); antonpuhovkin@gmail.com (A.P.); mishra.k@czechglobe.cz (K.B.M.); 503288@mail.muni.cz (D.G.); 2Fen Edebiyat Fakültesi, Biyoloji Bölümü (Department of Biology), Erciyes Üniversitesi (Erciyes University), 38039 Kayseri, Turkey; mghalici@gmail.com; 3Laboratory of Plant Ecophysiology and Climate Change, Environmental Sciences Department, Faculty of Natural Resources, Catholic University of Temuco, Avenida Rudecindo Ortega 02950, Campus San Juan Pablo II, Temuco 481 1123, Chile; angecasanova@gmail.com; 4State Institution National Antarctic Scientific Center, Ministry of Education and Science of Ukraine, T. Shevchenko blvrd. 16, 01601 Kyiv, Ukraine; 5Department of Reproductive System Cryobiology, Institute for Problems of Cryobiology and Cryomedicine, National Academy of Sciences of Ukraine, Pereyaslavska Str. 23, 61016 Kharkiv, Ukraine; 6Laboratory of Ecological Plant Physiology, Czech Academy of Sciences, Global Change Research Institute, Bělidla 4a, 603 00 Brno, Czech Republic

**Keywords:** non-photochemical quenching, photoinhibitory quenching, Antarctica, James Ross Island

## Abstract

The Antarctic lichen, *Xanthoria elegans*, in its hydrated state has several physiological mechanisms to cope with high light effects on the photosynthetic processes of its photobionts. We aim to investigate the changes in primary photochemical processes of photosystem II in response to a short-term photoinhibitory treatment. Several chlorophyll *a* fluorescence techniques: (1) slow Kautsky kinetics supplemented with quenching mechanism analysis; (2) light response curves of photosynthetic electron transport (ETR); and (3) response curves of non-photochemical quenching (NPQ) were used in order to evaluate the phenomenon of photoinhibition of photosynthesis and its consequent recovery. Our findings suggest that *X. elegans* copes well with short-term high light (HL) stress due to effective photoprotective mechanisms that are activated during the photoinhibitory treatment. The investigations of quenching mechanisms revealed that photoinhibitory quenching (qIt) was a major non-photochemical quenching in HL-treated *X. elegans*; qIt relaxed rapidly and returned to pre-photoinhibition levels after a 120 min recovery. We conclude that the Antarctic lichen species *X. elegans* exhibits a high degree of photoinhibition resistance and effective non-photochemical quenching mechanisms. This photoprotective mechanism may help it survive even repeated periods of high light during the early austral summer season, when lichens are moist and physiologically active.

## 1. Introduction

Photoinhibition of photosynthesis is a highly complex phenomenon that occurs due to light-induced loss of oxygen evolution and/or electron transport activities of photosystem II (PSII) (see Review [1]). Light-induced downregulation of photosynthesis, on the other hand, is called reversible or dynamic photoinhibition and can be caused by an increase in NPQ due to PSII reaction center closure in the presence of high light. In the chloroplast, light manifested a downregulation of both primary biophysical and secondary biochemical processes of photosynthesis, which also affects structural and functional changes in light harvesting complexes (LHCs) and photosystems I and II. Light-induced effects have been recently investigated in vascular plants (see [2]) and moist lichens (see [3]). It is accompanied by the production of harmful singlet oxygen (^1^O_2_) and other reactive oxygen species (ROS) in the thylakoid membrane of the chloroplast and PSII in particular (for review in lichens, see [4]). These ^1^O_2_ and ROS have a negative effect on proteins and pigment protein complexes that, in cases of severe photoinhibition, leads to their oxidative degradation and loss of function (see [5,6]). Photoinhibition-induced changes in PSII typically start with photooxidative damage of the D1 protein of the reaction center of PSII that must be repaired during the recovery period, which is energetically costly. The rate of D1 protein damage has been shown to be directly proportional to light intensity; this is a continuous phenomenon in which synthesis of damaged D1 protein continues at all light intensities. If the damage outpaces the synthesis, especially in bright light, non-functional PSII units accumulate. As a result, photosynthetic organisms, such as lichens, and their photosynthetizing partners have evolved several photoprotective mechanisms that dissipate excess excitation energy absorbed in chloroplasts via non-photochemical quenching (NPQ), which also serves as a protective pathway for lichen photosynthetic performance during desiccation stress (e.g., [7]) and low temperature stress [8]. Across the plant kingdom, NPQ and photoinhibition are strongly interdependent (for review, see [9]). According to the pathway of dissipation, three components of non-photochemical quenching are distinguished: energy-dependent quenching (qE), photoinhibitory quenching (qI), and state-transition quenching (qT) (for review, see e.g., [10]). In the initial phase of high light exposure, qE represents an important part of photoprotection since it is related to protonation of the lumen of the thylakoid (for algae, see e.g., [11]) and, consequently, to low pH-induced activation of violaxanthin de-epoxidase and formation of zeaxanthin, which serve as a quencher of excitation energy of PSII (xanthophyll cycle pigment mechanism—for lichens, see e.g., [12]). Apart from xanthophyll-cycle pigments, other quenchers and antioxidative compounds are effective in the photoprotection of lichens, such as glutathione [13]. State transition quenching (qT) is the component of non-photochemical quenching that balances energy absorption at low light between PSII and PS I by shifting LHCs. With progressive or severe photoinhibition, qI quenching takes a dominant role. It represents several structural changes in photosynthetic apparatus, such as the detachment of LHCs and the formation of LHC aggregates that consequently lead to the formation of quenching centers that reduce the negative effects of excess light. It was also shown to increase with cold acclimation in *Arabidopsis* natural accessions [14,15]. Recently, a two-step hypothesis of photoinhibiton has been proposed. It distinguishes the first phase (damage to the oxygen-evolving complex of PSII) followed by the second phase (traditional; see above of high light-induced damage destruction of PSII [16]. This hypothesis has been tested in lichens exposed to photoinhibitory treatment by light of different spectral qualities (UV light, blue light) [17].

Lichens activate the above-specified mechanisms during photoinhibition when they are in a physiologically active state, i.e., hydrated. However, photoinhibition-induced negative changes in PSII functioning have been reported even in the dry state of lichens [18]. In general, moist thalli of lichens are considered to be more susceptible to photoinhibition and PSII limitation because the cortex transmits more light when wet [19]. Moreover, lichen thalli showing a gradual loss of water during desiccation reduce the photochemical processes of photosynthesis associated with PSII due to (1) a desiccation-induced decrease in PSII effectivity and (2) effective quenching of excitation energy, which helps to protect PSII from the negative effects of photoinhibition as well.

Chlorophyll *a* fluorescence technique is a powerful and widely used method for the evaluation of the effects of biotic and abiotic stresses on underlying photosynthetic reactions, e.g., photoinhibition photosynthetic efficiency, photochemical and non-photochemical quenching mechanisms, on photosystem II (PSII). This method has been increasingly used to evaluate photoinhibition of primary photosynthetic processes in higher plants as well as in lichens in the last two decades. In chlorophyll fluorescence studies, changes in maximum quantum yield of PSII photochemistry (F_V_/F_M_) and effective quantum yield of PSII photochemistry (Φ_PSII_) during short-term photoinhibition and recovery have been used quite frequently in lichens (e.g., [20,21]). Quenching mechanisms and their components are studied much less frequently. Earlier studies addressing photoinhibition in Antarctic lichens focused mainly on the proportion of photoinhibitory quenching (qI) to energy-dependent quenching (qE), the two of the three components of non-photochemical quenching during a short-term photoinhibition induced by high light [22,23]. In addition, changes in glutathione contents in lichen thalli were investigated in response to photoinhibitory stress [24].

In this study, we used several below-specified chlorophyll fluorescence based methods to assess the response of the photosynthetic apparatus of *X. elegans* photobiont to high light treatment and activation of protective mechanisms, non-photochemical quenching in particular. Among the used chlorophyll fluorescence techniques, rapid light response curves (RLCs, *sensu* [25]) provide effective insights into photosynthetic processes, especially short- and long-term acclimation to light. The first aspect reflects the fact that typically only 20 s of exposure (or even shorter) to light is provided during RLC measurements [26]. The second aspect is useful in long-term acclimatory studies, in which plants are exposed to different light intensities for hours or days. The technique has therefore been used for the evaluation of photosynthetic acclimation in aquatic environments, e.g., sea grasses [27] grown in low light and those exposed to highlight intensities. Similarly, a study by Huang et al. [28] demonstrated that RLCs might be used for high-light acclimation studies in woody and fern species. Thus, RLCs represent a useful tool to study photoinhibition effects on photosynthetic processes. Recently, RLCs have been used mainly to study photoacclimation and photoinhibition in algae [29,30]. In lichens, only a few studies have used RLCs to monitor their response to micrometeorological factors, such as light and relative air humidity (e.g., [31] for Antarctic *Cladonia borealis*). To our best knowledge, no attempt has yet been made to evaluate the extent of photoinhibition and recovery in optimally hydrated lichens by the RLC. Thus, our study uses the RLC technique for the first time for the evaluation of photoinhibition under controlled conditions and discusses its potential for comparative (interspecific) studies of photoinhibition in lichens from well-lit environments.

Recently, another chlorophyll fluorescence method has been used for the assessment of the sensitivity of lichen photosynthetic processes to light: induction curves of non-photochemical quenching (NPQ) recorded on dark-adapted lichens when exposed to constant light for a certain period of time (typically tens of minutes). The merit of the method is a curvilinear (exponential) increase in NPQ with the time of exposure to light. The method was successfully applied in higher plants with different sensitivity to photoinhibition, e.g., a large variety of *Arabidopsis thaliana* mutants with modified structure and function of PSII, e.g., qE-deficient [32], LHC-deficient, with blocked violaxanthin synthesis [33], and in the absence of photosystem II subunit S (PsbS) [34]. Recently, the method was successfully used for the evaluation of species-specific differences in NPQ in shade lichens [4,35,36], as well as the differences between sun and shade ecotypes of individual species [36]. Last but not least, the method confirmed that the lichen *Crocodia aurata* can be hardened to photoinhibition by pretreatment with light [37]. Induction curves of the NPQ, however, have never been used for the assessment of lichen response to acute photoinhibition and the rapidity of recovery. Therefore, we used this method for the evaluation of immediate effects on NPQ in chlorolichen *X. elegans* treated by a short-term heavy photoinhibition (PAR of 2000 μmol m^−2^ s^−1^). We hypothesized that NPQ induction in response to medium light (100 μmol m^−2^ s^−1^) would result in higher NPQ values and a faster rate of induction kinetics in samples immediately after the photoinhibitory treatment than in the untreated control. In our study, we also looked at how the NPQ induction kinetics changed during the recovery period. We anticipated that the NPQ induction curve would show fast recovery towards the prephotoinhibition state.

## 2. Results

### 2.1. Photoinhibition-Induced Decrease in PSII Functioning and Consequent Recovery

Photoinhibitory treatment inhibited PSII processes, as evidenced by the changes in shape of the slow Kautsky kinetics (KK) of chlorophyll fluorescence (see Figure 1). The most remarkable change was a decrease in the overall chlorophyll fluorescence signal as a result of extensive photoinhibition-induced quenching. This caused a phenomenon of ‘flattening’ of the KK (cf. subplots A and B in Figure 1). After the photoinhibitory treatment, the maximum levels of chlorophyll fluorescence reached after the saturation pulse (F_M_, F′_M_ decreased in absolute values. (subplot B). During recovery (subplot C), both the chlorophyll fluorescence signal and the F_M_, F′_M_ values tended to increase toward pre-photoinhibition levels, showing *X. elegans* has a high potential to cope with negative effects caused by short-term photoinhibition.

Photoinhibitory treatment led to a significant decrease in the potential yield of photosynthetic processes in PSII (F_V_/F_M_) from 0.52 (before photoinhibition) to 0.28, which was followed by a swift increase to 0.50 during the consequent recovery phase (see Figure 2). The increase in F_V_/F_M_ during recovery was discovered to be biphasic, with a fast and a slow phase (for more details, see Discussion). However, the recovery of F_V_/F_M_ was not fully completed, and the PSII function was not fully recovered because its final value after 3 h of recovery was 95.9% of the initial (control) value. The effect of photoinhibitory treatment on the effective quantum yield of photochemical processes of photosynthesis in PSII (Φ_PSII_) was found to be similar to that of F_V_/F_M_. The Φ_PSII_ values decreased to 62.9% of the untreated control and then showed a biphasic increase towards the initial value. Recovery of Φ_PSII_ was fast and effective; however, it did not fully recover even after 3 h (98.3% of the initial value was reached after a 3 h recovery).

Non-photochemical quenching of chlorophyll fluorescence was activated by the photoinhibitory treatment, which is apparent from the NPQt time course. Immediately after the photoinhibitory treatment, NPQt increased by a factor of 2.2 (see Figure 3). The increase was followed by an exponential decrease in NPQt values recorded during the recovery period. At the end of a 3 h recovery period, NPQt did not reach its initial values, suggesting an uncompleted recovery. However, the photoinhibition-related component of non-photochemical quenching (qIt) played a dominant role in the overall quenching (see Figure 3), since qIt showed a similar time course and high values while energy-dependent quenching (qEt) remained low both in absolute values (close to 0), and in comparison with qIt.

### 2.2. NPQ Activation by Light in Control, Photoinhibited, and Recovered Thalli

In untreated *X. elegans*, NPQ increased curvilinearly with the time of exposition to light (PAR = 100 μmol m^−2^ s^−1^) and reached a maximum of 0.24 after 5 min of PAR (Figure 4). Immediately after photoinhibitory treatment, NPQ rose to a value of 0.44, documenting a photoinhibition-induced activation of NPQ processes that declined slightly with the time of exposition to PAR. During recovery time (30, 60, and 120 min), NPQ declined to the values of 0.30, 0.28, and 0.23 for the particular time of recovery. Upon exposure to PAR, NPQ values showed a curvilinear increase, with the final values of about 0.41 found at the end of the PAR period (5 min).

### 2.3. Photosynthetic Rapid Light Response Curves of Electron Transport

Photoinhibitory treatment led to a typical ‘flattening’ of the ETR light-response curve (see Appendix A), which is demonstrated by a photoinhibition-induced decrease in ETR_max_ values (see Figure 5). The minimum ETR_max_ value was found immediately after the photoinhibitory treatment, followed by a partial recovery at the end of the recovery period. The photoinhibitory treatment also had a negative effect on the shape of the initial part of the ETR light-response curve. The initial slope of the curve (α), interpreted as the effectivity of photosynthesis in low light, decreased immediately after the photoinhibitory treatment and remained low even during the first hour of the recovery period. Then, it showed an increase towards the initial values and an uncompleted recovery. This fact was reflected in an increase in the IK parameter, i.e., the minimum saturation irradiance calculated from the initial slope of the curve, which was apparent in response to the photoinhibitory treatment, followed by a decline with progressive recovery. F_V_/F_M_ x ETRmax/2 showed a substantial photoinhibition-induced decrease (by a factor of almost 3), followed by only a 50% recovery. This might be attributed to a partial limitation of photosynthetic processes in *X. elegans* even after 180 min of recovery.

## 3. Discussion

The chlorophyll fluorescence transient, also known as Kautsky kinetics (KK), is a very useful technique for probing photochemical activities; however, when combined with saturation pulses, it provides detailed information on photochemical and non-photochemical quenching processes that occurs in thylakoid membranes (in higher plants) or in the thalli of lichens. Photoinhibition-induced changes in Kautsky kinetics (KK) shape were found in the earlier study for the lichen *Usnea antarctica* [22]. Similarly to the study, an apparent ‘flattening’ of the KK was found immediately after photoinhibitory treatment. This was manifested by lower peak chlorophyll fluorescence (F_P_) and steady-state chlorophyll fluorescence (F_S_) levels see Figure 1. The interpretation of changes in the shape of the KK curve beyond F_P_ is highly complex due to the switch-on of several processes, e.g., changes in the acidification of the thylakoid lumen, ATP synthesis and activation of the Calvin-Benson cycle, and structural and functional properties of the thallus. These findings might be attributed to strong photoinhibition, as evidenced by the fact that initial (F_O_) chlorophyll fluorescence was found to be lower in photoinhibited than control (before), which is a typical response to heavy photoinhibition in lichens [38,39]. Another fact supporting the idea of strong photoinhibition induced by the experimental treatment is that the chlorophyll fluorescence signal showed a slightly increasing trend in KK (while actinic light was switched on—see an arrow in Figure 1, subplot B) and even increased when the actinic light was switched off (marked with an asterisk). This suggests that LHCs were inhibited (via photoinhibitory treatment) and overenergized by the actinic light, causing them to emit more chlorophyll fluorescence in the dark period than during the exposition to actinic light.

Time courses of F_V_/F_M_ and Φ_PSII_ showed similar shapes to those recorded for a variety of photoinhibited lichen species [3,17,40], i.e., a photoinhibition-induced dip with consequent recovery, documenting a light-induced decrease in the effectivity of PSII functioning followed by a well-distinct fast (within the first 30 min recovery) and slow phase (over 30 min) of recovery, similarly to the earlier evidence reported for Antarctic lichen species [41]. The almost complete recovery of F_V_/F_M_ and Φ_PSII_ (Figure 2) demonstrated a high capacity of *X. elegans* to cope with short-term strong photoinhibition (evaluated at a temperature of 5 °C). In the field, however, interactions with even lower temperatures occur, which brings about the phenomenon of low-temperature-induced photoinhibition of photosystem I (PSI), which has not yet been investigated in lichens. Therefore, its contributions to the limitations of primary photosynthetic processes are still unknown. However, the interaction of low temperatures and photoinbibition in Antarctic lichens is of great importance. The studies focused on the temperature response curves of primary photosynthesis report that a significant drop in F_V_/F_M_ and Φ_PSII_ occurs when the temperature decreases from +10 to −15 °C [42,43]. Thus, when considering the photosynthetic adaptation of Antarctic lichens to photoinhibition in field conditions, the combination of low temperature and high light-induced decreases in PSII functioning must be considered. Such analysis could help in predictions of the ecophysiological behavior of lichens in the field as well as their ability to withstand low-temperature photoinhibition [44].

The strong photoinhibitory treatment activated photoinhibitory quenching (qIt), which simultaneously decreased F_V_/F_M_ and Φ_PSII_ (see the increase in qI values in Figure 3), and it relaxed rapidly during the early recovery period. The energy-dependent quenching qEt was close to zero. This is comparable to the study using the same methods of photoinhibitory treatment and analysis [41] that found no involvement of qE in *Umbilicatia decussata* passing photoinhibition and recovery. However, the same study discovered the involvement of qE in *Usnea antarctica*. The difference could be attributed to the morphotype of the thalli because foliar microlichens are exposed to high light more or less evenly over the thallus area (both in the field and in a laboratory), whereas fruticose lichens, due to their complex branching structure and intershading effect, may have differently lit thalli parts. As a result, some thalli parts may experience less than maximum irradiation and, consequently, less severe photoinhibition. In such a situation, the qE component of non-photochemical quenching may be useful.

Induction curves of NPQ, recorded at constant light, in *X. elegans* thalli that had undergone photoinhibitory treatment and subsequent recovery revealed that NPQ increased by a factor of about 1.8 when compared with the untreated control (Figure 4). The control induction curve was exponential, similar to those reported for several lichen species [36,45]. However, the NPQ induction curves recorded in our study for photoinhibited *X. elegans* had a different shape. The photoinhibition-induced increase in NPQ was most noticeable during the first 60 s of experimental light on. This means that even low light intensity and its short duration caused stress in PSII, as evidenced by a rapid NPQ increase to a maximum value of 0.5 in the thalli measured immediately after photoinhibitory treatment. Involvement of non-photochemical quenching decreased with time of recovery, which was shown as a less steep increase in NPQ values at the beginning of the induction curve with the time of recovery and a gradual return of the shape of the induction curve to control (see the curve recorded after 120 min of recovery).

Rapid light response curves of ETR were used to evaluate the effects of photoinhibition in hydrated lichens for the first time. The method was found effective in the evaluation of photoinhibition since significant changes were apparent both in the shape of the ETR curves (see Appendix A) and in the parameters derived from the ETR curves. Photoinhibition-induced decrease in α, ETR_max_, F_V_/F_M_ x ETR/2 was found. Typically, these parameters increase with an increase in light in the physiological range in the order of low, moderate, and high light [46]. Severe photoinhibition, on the other hand, causes a decrease in the parameters due to increased NPQ and less effective PSII functioning immediately after photoinhibitory treatment. Similar responses, i.e., a decline in α, ETR_max_, F_V_/F_M_ x ETR factor/2, were observed for other stressors, such as high temperatures [47]. In our study, the three parameters showed photoinhibition-induced decreases and significant recovery, implying that the method of ETR curves could be used to indicate photoinhibition’s short-term changes. Therefore, it might be recommended that follow-up studies of primary photosynthetic processes in lichen focus on photoinhibition effects carried out both in the field and in laboratory conditions.

## 4. Material and Methods

### 4.1. Collection and Handling of Samples

The samples of *X. elegans* were collected from a deglaciated area of the northern part of James Ross Island during the Czech Antarctic expedition during the austral summer season 2021–2022. The locality of the collection is a slightly inclined plateau (50 m a.s.l.) close to a coastal line located north of the Berry Hill mesa (63.806293 S, 57.842755 W; see Figure 6). Lichen thalli were collected from stone surfaces along a shallow stream (Figure 7), described earlier as an area with a high biodiversity of lichens [48]. After collection, the thalli were naturally dried in a shaded, windy place close to the J.G. Mendel station. Dried samples were stored and transferred to the Extreme Life Laboratory of Masaryk University, Brno, where they were stored in a refrigerator at a temperature of 5 °C.

Before experiments, the thalli of *X. elegans* were rewetted for 24 h under dim light (20 μmol m^−2^ s^−1^ PAR) at a temperature of 5 °C. Before the photoinhibitory treatment, the samples were tested for the potential yield of the photochemical processes of photosynthesis (F_V_/F_M_). When F_V_/F_M_ reached a constant value, the primary photosynthetic processes were considered fully activated by rehydration. The thalli were then used for the experiments studying different aspects of *X. elegans* sensitivity to photoinhibition and activation of protective mechanisms, components of non-photochemical quenching in particular.

### 4.2. Photoinhibitory Treatment

Unless otherwise stated, thalli of *X. elegans* were treated with 2000 μmol m^−2^ s^−1^ of photosynthetically active radiation (PAR) for 30 min. During the photoinhibitory treatment as well as the consequent recovery, thallus temperature was maintained at 5 °C (monitored by a needle CuCo termocouple linked to a datalogger EdgeBox V12 (Environmental Monitoring Systems, Brno, Czech Republic)) by a jacket cooling unit filled with ice. The same temperature was kept during chlorophyll fluorescence measurements (see below).

### 4.3. Photoinhibition and Recovery of PSII

Before (control), after the photoinhibitory treatment, and during the recovery period, chlorophyll fluorescence measurements were made using a HFC-010 fluorometer and FluorCam07 software (Photon Systems Instruments, Brno, Czech Republic). To evaluate the effect of photoinhibitory treatment, slow Kautsky kinetics (KK) supplemented with quenching analysis were used, exploiting the method optimized for lichens (see [49]). Parameters derived from KK are specified in Table 1. A single measurement started with a saturation pulse applied in the dark-adapted state of the lichen thallus to induce maximum chlorophyll fluorescence level (F_M_), followed by a short 10 s dark period. Then, the lichen samples were exposed to actinic light (AL = 4) for 300 s, and a slow chlorophyll fluorescence curve was recorded. When a steady-state ChlF was reached (typically after 300 s), a saturation pulse was applied to induce F′_M_ (maximum chlorophyll fluorescence) in the light-adapted state. After switching off the actinic light, background ChlF (F′_O_) was recorded for 30/60/90 s. Then, other repetitive saturation pulses were given in order to induce F″_M_ levels in the dark. Then, the following chlorophyll fluorescence parameters were calculated by FluorCam software:

Additionally, quenching parameters were calculated from raw data using the equation by [53] for the evaluation of photoinhibitory quenching. For the calculation of light-adapted minimum fluorescence (F′_O_), the Oxborough method was used [54].

Using the data of the above-specified chlorophyll fluorescence parameters, time courses were constructed, and the parameters evaluating photoinhibition-induced changes and their recovery (at low light of 5 μmol m^−2^ s^−1^) were evaluated. There was a relative decline or increase in F_V_/F_M_ and Φ_PSII_, NPQ, qIt, and qEt, as well as a fast and slow recovery of these parameters.

### 4.4. Induction Kinetics of NPQ

Induction kinetics were measured according to a chlorophyll fluorescence protocol (modified [4]) using a PAM 2500 fluorometer (Heinz Walz GmbH, Germany). To determine the induction kinetics of NPQ, *X. elegans* thalli (control, those exposed to photoinhibitory treatment, and those after 30, 60, and 120 min recovery) were dark-adapted for 10 min and then exposed to constant actinic light of 100 μmol (photons) m^−2^ s^−1^ for 10 min. During the period, repetitive saturating pulses were applied in order to evaluate the NPQ. The NPQ values were calculated using the formula [52]: NPQ = (F_M_ − F′_M_)/F_M_

### 4.5. Rapid Light Response Curves of Photosynthetic Electron Transport

A pulse-amplitude modulated fluorometer (PAM 2500, Heinz Walz GmbH, Germany) and the protocol measuring light-response curves were used for the measurements of photosynthetic electron transport rate (ETR). Prior to the chlorophyll fluorescence measurement, thalli of *X. elegans* were predarkened for 10 min in darkening lips (the time long enough to fully reoxidize PSIIs—as tested before). Then, the thalli (5 replicates) were exposed to photon flux densities (PFDs) of 0, 3, 7, 32, 65, 105, 142, and 200 µmol (photons) m^−2^ s^−1^ of photosynthetically active radiation (red light) for 20 s at each light level. After the 20 s light exposure, a saturating pulse was applied to the sample to detect the effective photochemical quantum yield of photosystem II (PS II): Φ_PSII_ = (F_M_′–F_S_)/F_M_′ [51]. The (photosynthetic) electron transport rate of PSII (ETR) was calculated using the equation:ETR = Φ_PSII_ × PFD × absorption/2
where Φ_PSII_ is the effective PSII quantum yield, PFD = photon flux density, and absorption was considered 0.84, reflecting the fraction of incident photons absorbed by the chlorophyll a molecules in PSII. Division by 2 due to the equal fraction of photons absorbed by PSII and PSI.

Rapid light-response curves were constructed by PAM software by plotting the electron transport rate (ETR) versus the increasing light (from 1 to 1 300 μmol m^−2^ s^−1^) with intervals of 20 s between particular light levels. The best fit was performed using the method of [55]. Photosynthetic parameters (see a list in Table 2) were derived from the ETR light response curves.

### 4.6. Statistical Analysis

Statistically significant differences in the chlorophyll fluorescence parameters were evaluated by one-way ANOVA (Fisher LSD test, *p* < 0.05) in the Statistica vs. 14 package (TIBCO Software Inc., Palo Alto, CA, USA).

## 5. Concluding Remarks

We have used chlorophyll fluorescence techniques with a variety of laboratory experiment designs to investigate the phenomenon of photoinhibition in an Antarctic lichen species, *Xanthoria elegans*. Analysis of non-photochemical quenching could provide insights into the species ability to adapt its photosynthetic performance to high light. This is especially true for maritime Antarctic environments during the early Austral summer season, when moisture availability is typically high and short-term high light episodes may occur frequently. The short-term changes in PSII photochemistry observed in our study support the idea that *X. elegans* is well adapted to such short-term photoinhibitory episodes, particularly by its ability to rapidly increase NPQ, and non-photochemical quenching component related to photoinhibition.

## Figures and Tables

**Figure 1 plants-12-02259-f001:**
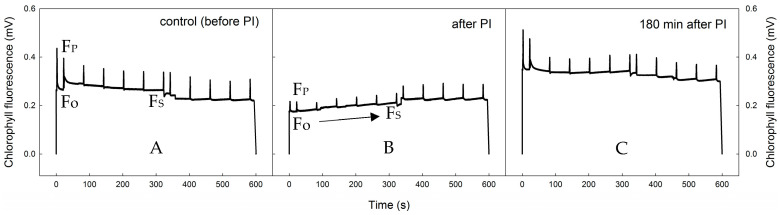
Records of slow Kautsky kinetics of chlorophyll fluorescence (means of nine replicates) taken before (**left**, (**A**)), immediately after the photoinhibitory treatment (**central**, (**B**)), and after 180 min recovery (**right**, (**C**)) in wet thalli of *X. elegans*. The arrow indicates increasing chlorophyll fluorescence values during the period of actinic light on Fo is an initial chlorophyll fluorescence induced by low light, F_P_ is a peak chlorophyll fluorescence reached when experimental (actinic) light is switched on, F_S_ is a steady-state chlorophyll fluo-rescence after a 5 min acclimation to experimental light.

**Figure 2 plants-12-02259-f002:**
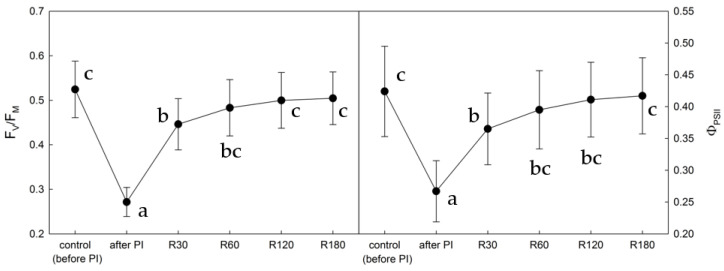
Time courses of potential yield of photosynthetic processes in PSII (F_V_/F_M_), effective quantum yield of photosynthetic processes in PSII (Φ_PSII_) recorded in *X. elegans* before, immediately after the photoinhibitory treatment (after PI), and after 30, 60, 120, and 180 min recovery (R30, R60, R120, and R180). Data points represent the means of five replicates ± standard deviations (error bars). The lower cases indicate statistically significant difference at *p* < 0.05.

**Figure 3 plants-12-02259-f003:**
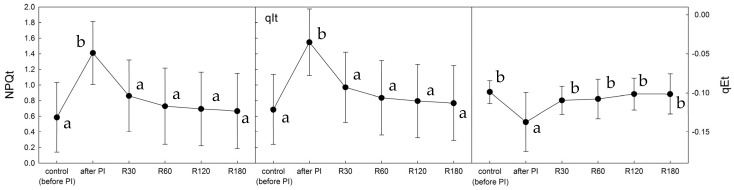
Time courses of non-photochemical quenching (NPQt—**left**), photoinhitory- and energy-dependent quenching of chlorophyll fluorescence (qIt, qEt—**central**, **right**) showing involvement of photoprotective mechanisms during photoinhibitory treatment and subsequent recovery. The NPQt, qIt, qEt values were evaluated before (control), immediately after the photoinhibitory treatment (after PI), and after 30, 60, 120, and 180 min recovery (R30, R60, R120, and R180). Data points represent the means of five replicates ± standard deviations (error bars). The lower cases indicate statistically significant difference at *p* < 0.05.

**Figure 4 plants-12-02259-f004:**
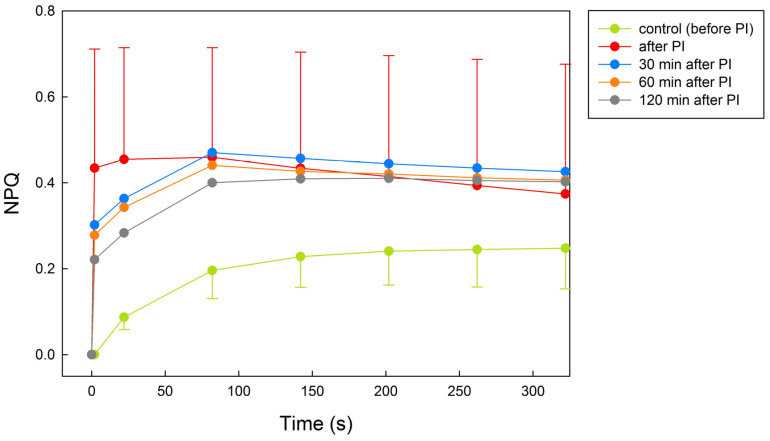
Time courses of NPQ (induction kinetics) in response to constant PAR (see below) recorded before, immediately after the photoinhibitory treatment (PI), and after 30, 60, and 120 min recovery. During the NPQ measurements, *X. elegans* thalli were exposed to PAR of 100 μmol m^−2^ s^−1^ lasting for 5 min. Data points represent the means of five replicates. Standard deviations (error bars) are shown only for the untreated control and the thalli measured immediately after photoinhibitory treatment. The NPQ values are statistically significantly different between control and after PI treatment (not shown, lower cases are missing).

**Figure 5 plants-12-02259-f005:**
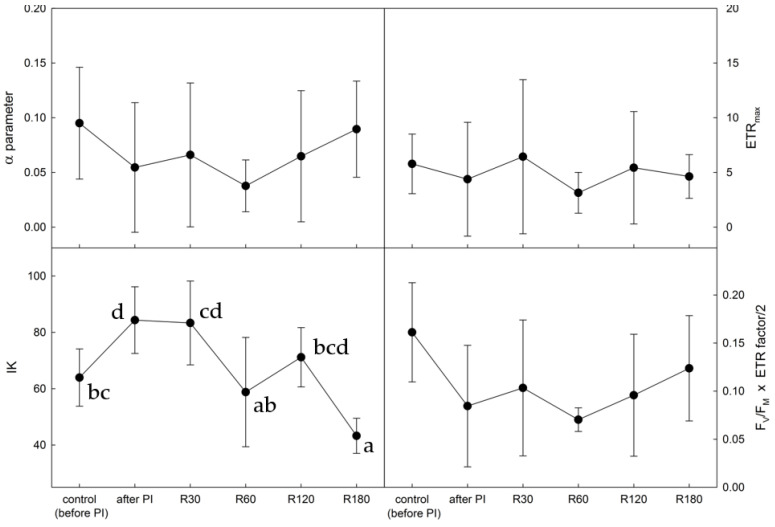
Time courses of photosynthetic parameters derived from rapid light-response curves of photosynthesis (see Appendix A) in *Xanthoria elegans* recorded before photoinhibitory treatment (control), immediately after (after PI), and during recovery from the photoinhibitory treatment (R30, R60, R120, and R180—for definition, see Figure 2 legend). Data points represent the means of five replicates ± standard deviation (error bars). The lower cases indicate statistically significant difference at *p* < 0.05. In other panels, there is no statistically significant difference. Therefore, no lower cases are added.

**Figure 6 plants-12-02259-f006:**
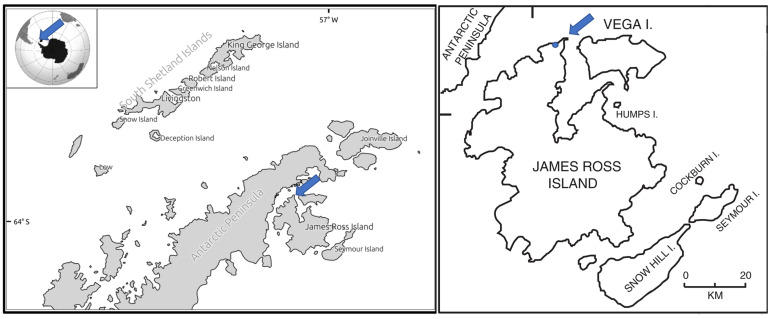
Geographical location of James Ross Island, Antarctica, with indication of collection site.

**Figure 7 plants-12-02259-f007:**
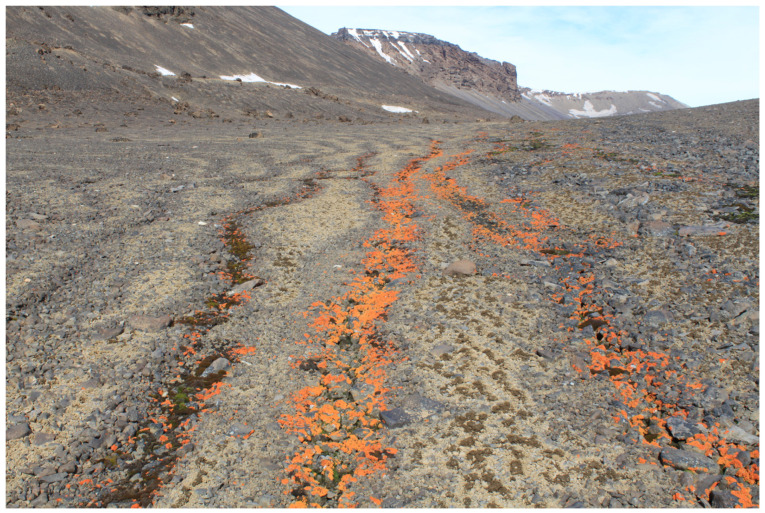
Locality of sampling of *X. elegans* on the northern part of James Ross Island (Antarctica). Together with mosses, *X. elegans* forms well-distinguished lines, filling the depressions formed on surfaces with slight inclines.

**Table 1 plants-12-02259-t001:** List of chlorophyll fluorescence parameters derived from slow Kautsky kinetics supplemented with quenching analysis.

Parameter	Meaning	Source
F_V_/F_M_	maximum (potential) yield of photosynthetic processes in PSII	[50]
Φ_PSII_	effective quantum yield of photosynthetic reactions in PSII	[51]
NPQ	non-photochemical quenching *according to*	[52]
NPQt	non-photochemical quenching *according to*	[53]
qIt	photoinhibitory quenching	[53]
qEt	energy-dependent quenching	[53]

**Table 2 plants-12-02259-t002:** Definition of the ETR light response curve-derived photosynthetic parameters used in this study.

Parameter	Meaning
α	maximum quantum yield for whole-chain electron transport (“alpha” represents the slope of the ETR curve at low light intensities)
IK	PAR value of the point of intersection between a horizontal line ETR_max_ and the extrapolated initial slope.
ETR_max_	maximum electron transport capacity found at light saturation
F_V_/F_M_ x ETR factor/2	maximum yield of electron transport calculated from F_V_/F_M_

## Data Availability

Not applicable.

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
