# Peer review of "Resistance of Primary Photosynthesis to Photoinhibition in Antarctic Lichen Xanthoria elegans: Photoprotective Mechanisms Activated during a Short Period of High Light Stress"

_plants, 2023, doi:10.3390/plants12122259_

Round 1

Reviewer 1 Report

The authors investigated the photosystem II and photoportective responses in Antarchric lichen to short-term high light treatment using several ways. With the detailed introduction of methods, appropriate presentation of data and data interpretation, it provides significant information of the photoprotective mechanisms during short-term high light stress. While there are some minor concerns before accepting:

1) line 172-178, why the thalli were rewetted under low temperatures, and why this study is completed under low temperature stress, in which the constant value of Fv/Fm was about 0.5, showing the samples were under stress? can you provide the data of rewetting?

2) What is the light intensity of dim light (line 172)?

3) There is no data of significant differences (p<0.05) in figures, like Fig.4 and 5.

4) There are places authors used the word 'indicate", which is too strong. 

Author Response

Reviewer 1

Item                                                                     Answer

Why the thalli were rewetted at 5oC?     It is a standard method used in our lab when rewetting Antarctic lichens to mimick natural processes in the field. The temperature of 5 °C does not bring a decrease in Fv/Fm. The initial value of 0.5 was constitutive in the samples. In           pre-tests, it was more or less constant in the temperature range of +5 to +19 °C.

What was dim light intensity?                   We added the value of 20 mmols m-2 s-1 into the text.

In Figs., statistical significance is missing.             We added indication of statistical significance (a,b,c,d) into Figs.

´To indicate´ is used to frequently in the text.    We changed where appropriate.

Reviewer 2 Report

Dear Authors and Editors

The article „Resistance of primary photosynthesis to photoinhibition in Antractic lichen Xanthoria elegans: Photoprotective mechanisms activated during a short-period of high light stress ” was written in a transparent manner. The study is interesting in its assumptions.

The topics presented in the article are appropriate for the Journal's profile.

The literature is well selected.

The conclusions sum up the entire article appropriately.

I have only one technical note:

Line 189 - The presented temperature measurement result should have the same number of significant figures as the standard deviation.

In my opinion, this paper can be accepted for publication in Plants after major corrects.

Best regards

Author Response

Reviewer 2

Item                                                                     Answer

The temperature should have the same number of significant figures.

We rejected the ± because the value was not standard deviation but the difference to the extreme value reached during temperature monitoring (only once during the measurements).

Mean of 5 °C was left in the text.

Reviewer 3 Report

Dear Authors,

The present study evaluates the changes in primary photochemical processes of photosystem II in response to a short-term photoinhibitory treatment. The research subject is interesting and brings scientific important data in the field. Some changes of the manuscript should nevertheless be performed in order to improve its quality. Following specific changes should thus be performed:

Major changes

Introduction: You do not need to specify when citing a reference “see… [reference]. Line 121: … a study by [28]… needs to be replaced by the name of the first author.

Materials and Methods: You do not offer references for neither of the assays. Are all methods completely new? Because, if they are, you need to offer them a different approach. Please add numbers for the sub-chapters of this section, as recommended in the Instructions for Authors.

Results should also have numbers for sub-chapters, as recommended in the Instructions for Authors.

Discussions: Novelty and originality of your study should be emphasized in this section once again. You need to emphasize this in terms of results, not purposes, as the Introduction should.

Please change order of sections, as recommended in the Instructions for Authors. Please place the figures at the end of the manuscript, in the right place of the manuscript, as recommended in the Instructions for Authors.

Conclusions: Please add perspectives of your study.

All these suggested changes should be performed in order to bring further improvements to the manuscript. 

English language is fine, only a minor spell check is needed.

Author Response

Reviewer 3

Item                                                                     Answer

Reference line 121 (put the name of the first author)                                    Done.

Add number of subchapters into Material and Methods                                              Done

                                                                                              Citation of Oxborough was added.

Add subchapter numbers into Results                                                                   Done

Novelty should be emphasized.                               We did it for ETR curves in photoinhition in lichens. We added several lines.

Change the order of manuscript sections.            We moved Material and Methods to required place

English should be checked.                                        We made several improvements according to a native English speaker who checked the text.
